# Clopidogrel as a donor probe and thioenol derivatives as flexible promoieties for enabling H$_2$S biomedicine

Yaoqiu Zhu[1], Elkin L. Romero[1], Xiaodong Ren[1], Angel J. Sanca[1], Congkuo Du[2], Cai Liu[3], Zubair A. Karim[4], Fatima Z. Alshbool[4], Fadi T. Khasawneh[4], Jiang Zhou [5], Dafang Zhong[3] & Bin Geng[2]

Hydrogen sulfide has emerged as a critical endogenous signaling transmitter and a potentially versatile therapeutic agent. The key challenges in this field include the lack of approved hydrogen sulfide-releasing probes for in human exploration and the lack of controllable hydrogen sulfide promoieties that can be flexibly installed for therapeutics development. Here we report the identification of the widely used antithrombotic drug clopidogrel as a clinical hydrogen sulfide donor. Clopidogrel is metabolized in patients to form a circulating metabolite that contains a thioenol substructure, which is found to undergo spontaneous degradation to release hydrogen sulfide. Model studies demonstrate that thioenol derivatives are a class of controllable promoieties that can be conveniently installed on a minimal structure of ketone with an α-hydrogen. These results can provide chemical tools for advancing hydrogen sulfide biomedical research as well as developing hydrogen sulfide-releasing drugs.

[1] Department of Chemistry and Biochemistry, Border Biomedical Research Center, The University of Texas at El Paso, El Paso, TX 79968, USA. [2] Hypertension Center, Fuwai Hospital, CAMS-PUMC, State Key Laboratory of Cardiovascular Disease, Beijing 102300, China. [3] State Key Laboratory of Drug Research, Shanghai Institute of Materia Medica, Chinese Academy of Sciences, 501 Haike Road, Shanghai 201203, China. [4] Department of Pharmaceutical Sciences, School of Pharmacy, The University of Texas at El Paso, El Paso, TX 79902, USA. [5] Analytical Instrumentation Center, College of Chemistry and Molecular Engineering, Peking University, Beijing 100871, China. Correspondence and requests for materials should be addressed to Y.Z. (email: yzhu2@utep.edu)

                                                    1

Known as a stinky and noxious gas, hydrogen sulfide ($H_2S$) has emerged as an important gasotransmitter that mediates a myriad of physiological and pathological processes in human bodies[1–4]. The chemical biology of $H_2S$ are conveyed by three categories of reactions: (1) binding to the metal centers of proteins, (2) modulation of free radicals, and (3) modification of protein cysteines to persulfides[4]. Since the original discovery was unveiled in 1996, great effort has been made to explore the mechanisms of $H_2S$ regulation, yet many details remain unclear. One major challenge is that the $H_2S$ homeostasis in the human body is hard to alter for biomedical exploration. This is because $H_2S$ is produced from the natural substrates L-cysteine or L-homocysteine under a stringent enzymatic system[5–8]. Although some plant-derived substances such as polysulfides in garlic have been found to release $H_2S$, dietary intake represents a limited route for pharmacological intervention[9–11] So far, all $H_2S$-donating agents have been examined only in preclinical or early clinical studies, and in human exploration awaits the first donor probe to be approved[12–14].

The versatile roles of $H_2S$ in biological regulations have made it a potentially potent therapeutic agent for treating many human diseases[15–17]. This drug discovery direction has attracted increasing attention in recent years, and $H_2S$-releasing derivatives of simple organic compounds or known drugs (i.e., vehicle molecules) are under rapid development for augmenting the intriguing $H_2S$ pharmacology under pathological states. In order to manifest the therapeutic benefits and mitigate the potential toxicities of $H_2S$, an ideal donor needs to be activated in a selective manner in response to certain stimuli for controlling the gasotransmitter level under deleterious threshold, in a way similar to the enzyme-mediated endogenous production[5–8]. Although many synthetic donors have been reported, most of them undergo spontaneous hydrolysis to release $H_2S$ in an uncontrollable manner, which compromises their therapeutic potentials. In recent years, $H_2S$ promoieties such as acylated N–S[18–20] and S–S derivatives[21,22], caged-carbonyl sulfides[23–28], and iminothioethers[29,30] have been reported, and they require more specific bioactivation, e.g., thiol attack, for the $H_2S$ release. However, the chemical complexity of these precursors limits their versatility in assembly with different vehicle molecules. To develop diversified $H_2S$ donors of controllable release, precursors that can be selectively activated, conveniently synthesized and flexibly installed on a wide spectrum of chemical scaffolds are highly desired[31].

Herein, we report our recent discovery of a hidden $H_2S$-releasing pathway in clopidogrel (CPG) bioactivation, which establishes this widely used antithrombotic drug to be a $H_2S$-donating agent in clinic. Spawning from the $H_2S$-releasing thioenol substructure in CPG bioactivation, a strategy is formulated to derivatize the thioenol tautomer of thioketone into $H_2S$ promoieties, which can be assembled onto a minimal vehicle structure of ketone with an α-hydrogen (enolizable ketone). Model studies demonstrate that masked thioenols are a class of $H_2S$ donors with high installation flexibility and bioactivation selectivity.

## Results

**Identification of CPG as a clinical $H_2S$ donor**. The discovery stems from our recent bioactivation studies of CPG. Since being launched in 1997, the prevalent CPG treatment has been associated with unpredictable clinical outcomes including high intersubject variability[32,33]. CPG is an antithrombotic prodrug, and its thiophene moiety undergoes extensive metabolism including cytochrome P450s (CYPs)-catalyzed oxidation and paraoxonase-1 (PON-1)-catalyzed hydrolysis in patients' liver to form the thiol-containing active metabolite M13 (H3 and H4) and its endo isomer M15 (Fig. 1)[34–41]. In clinic, upon oral administration, plasma samples are treated with derivatization reagent 3′-methoxyphenacyl bromide (MP-Br), and the three circulating thiol metabolites, i.e., M13-H3, M13-H4 and M15, are measured by LC-MS/MS as their stabilized phenacyl derivatives (Fig. 1). Although H4, not H3, has demonstrated antiplatelet activity in vitro[34,36], its metabolic activation has shown only partial correlation with the observed clinical outcomes, which implicates that CPG might form additional active metabolites[40,42,43]. On the other hand, although PON-1 was found to be involved in M15 formation and has shown to be a genetic determinant of CPG responsiveness in certain patient populations[44–46], M15 has been considered as an inactive circulating metabolite. This remains a vigorously debated controversy in CPG clinical pharmacology.

The obscure roles of these circulating CPG metabolites can be attributable to the lack of synthetic standards for pharmacological evaluations. In an effort of identifying additional active metabolites of CPG to unravel its clinical puzzle, we have chemically synthesized M15 in its disulfide form, M15-DS (Fig. 2). The disulfide was prepared from a reflux reaction of synthetic M2 in methanol and toluene (1/4, v/v) at 100 °C followed by a selective hydrolysis (Fig. 2)[47]. M2 is the stable bioactivation intermediate and can be synthesized conveniently[48]. It has an α,β-unsaturated thiolactone structure, in which the carbonyl functional group is conjugated with a double bond and stabilized by a S-atom. The thioester carbonyl group in M2 is not active for nucleophilic attack until after a heat-promoted double-bond isomerization takes place in the five-member ring (M2 to M2'). This exo-to-endo migration not only cancels the double-bond conjugation with the carbonyl group but also switches the S-atom to a good leaving group in a thioenol form. Upon a nucleophilic attack by MeOH, the thioenol (M15-OMe) formed from M2' can

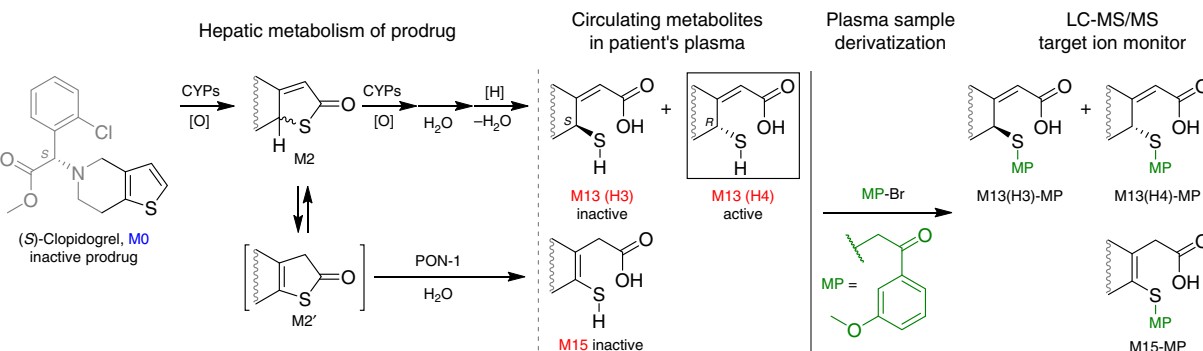

**Fig. 1** Metabolic activation of CPG in patients. The prodrug forms three circulating thiol metabolites (M13-H3, M13-H4 and M15) after hepatic metabolism; these reactive metabolites are converted to stabilized derivatives in clinical monitoring

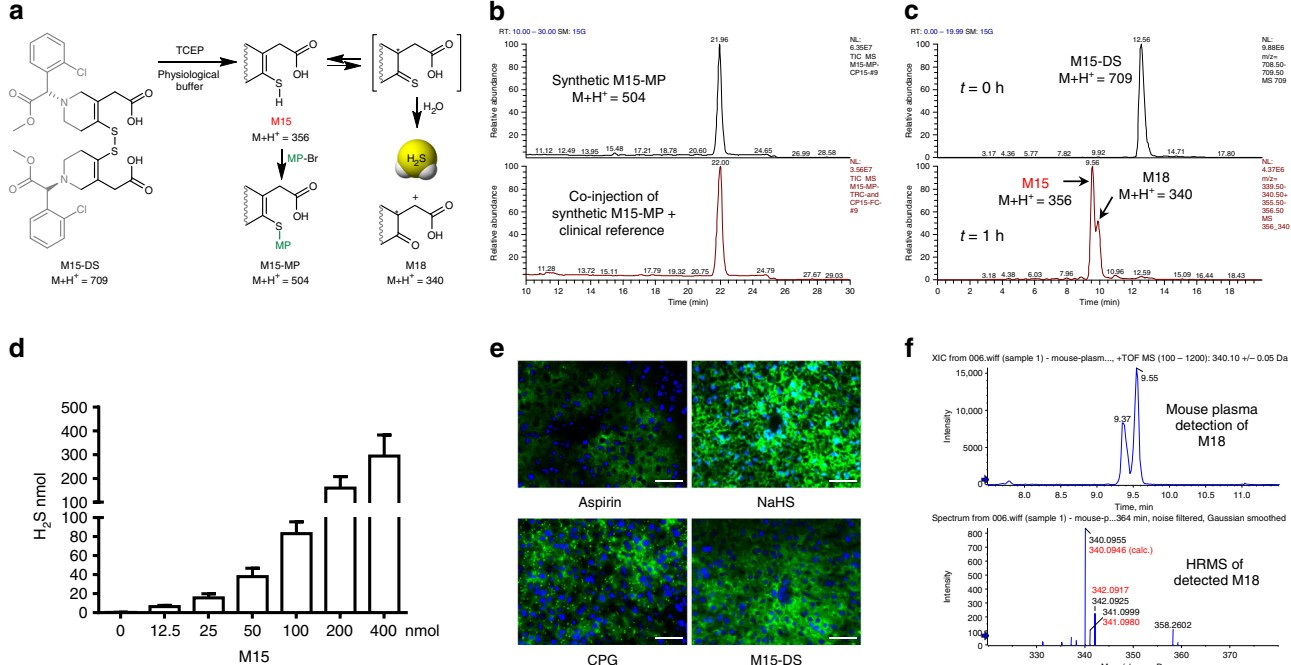

**Fig. 2** Chemical preparation of M15 disulfide (M15-DS) from the stable bioactivation intermediate M2. The heat-promoted double-bond migration in M2 is followed by thiolactone cleavage and disulfide formation

**Fig. 3** Human circulating metabolite M15 is a $H_2S$ donor. **a** Reductive cleavage of synthetic M15-DS in vitro to M15 followed by chemical derivatization or spontaneous degradation. **b** Synthetic M15-MP shows to be identical to clinical reference. **c** M15 conversion to desulfurized metabolite M18 in vitro; **d** stoichiometric $H_2S$ release from M15 detected by methylene blue method in vitro (data is displayed as means ± S.D., $n = 4$). **e** In vivo mouse studies of $H_2S$ release from CPG and M15-DS using fluorescent imaging probe Mito-HS ($n = 4$). Scale bars are 50 μm. **f** Detection of $H_2S$-released metabolite M18 in mouse plasma from in vivo studies

quickly be oxidized by air to its disulfide form (M15-OMe-DS), and $^1H/^{13}C$/DEPT135 NMR studies have confirmed the endo double-bond structure (Supplementary Fig. 1 and Supplementary Fig. 5). Upon treatment with concentrated HCl at room temperature, M15-OMe-DS undergoes selective methyl ester hydrolysis to yield M15-DS (Fig. 2).

M15-DS was found to be stable, and can be quickly converted to M15 in physiological buffer upon disulfide cleavage by bioreductive agents such as tris(2-carboxyethyl)phosphine (TCEP) (Fig. 3a). The formed M15 was trapped by MP-Br, and this synthetic M15-MP was confirmed to be identical to its clinical reference under LC-MS/MS studies (Fig. 1b and Supplementary Fig. 2). In physiological buffer, upon reductive release, M15 has been found to undergo spontaneous hydrolysis to release $H_2S$ and form a desulfurized product M18 (Fig. 3c), possibly through an equilibrium with the thioketone tautomer followed by hydrolysis (Fig. 1a). The $H_2S$ released from M15 was detected by traditional methylene blue method[49], and the results have demonstrated stoichiometric release of $H_2S$ (Fig. 3d). These in vitro studies support that the major circulating metabolite of CPG, M15, is a facile $H_2S$ donor. This $H_2S$ release pathway was then tested in vivo in mice. Upon intraperitoneal administration of M15-DS or CPG itself, the exogenous $H_2S$ released from the dosages has been trapped by fluorescent probe Mito-HS and imaged by laser-scanning confocal microscopy (Fig. 3e)[50,51]; the $H_2S$-released product of M15, M18, has been detected in the corresponding mouse plasma samples by UPLC-MS/MS (Fig. 3f). To test if $H_2S$ is released from M15 in human bodies, we conducted studies on pooled plasma samples collected from six healthy Chinese volunteers, 1 h after they took CPG (300 mg). The collected plasma samples were treated with standard clinical monitoring procedures including derivatization by MP-Br. Under UPLC-MS/MS analyses, in addition to M15-MP, the desulfurized metabolite M18, has also been detected (Fig. 4a, b). The ketone substructure in M18 was found to undergo further bioreduction to form a secondary alcohol metabolite, M18H, which is also a major circulating metabolite detected in the human plasma samples (Fig. 4a, c). The detected metabolites have shown to be identical to the synthetic standards (Fig. 4 and Supplementary Fig. 3). These human studies support that CPG is a clinical $H_2S$ donor.

As a potent regulator in cardiovascular systems, $H_2S$ and its donors have demonstrated various protective effects including antithrombosis and vasodilation in vitro and in animal studies[52-56]. Although M15 has long been detected as a circulating metabolite, the chemical stabilization in clinical

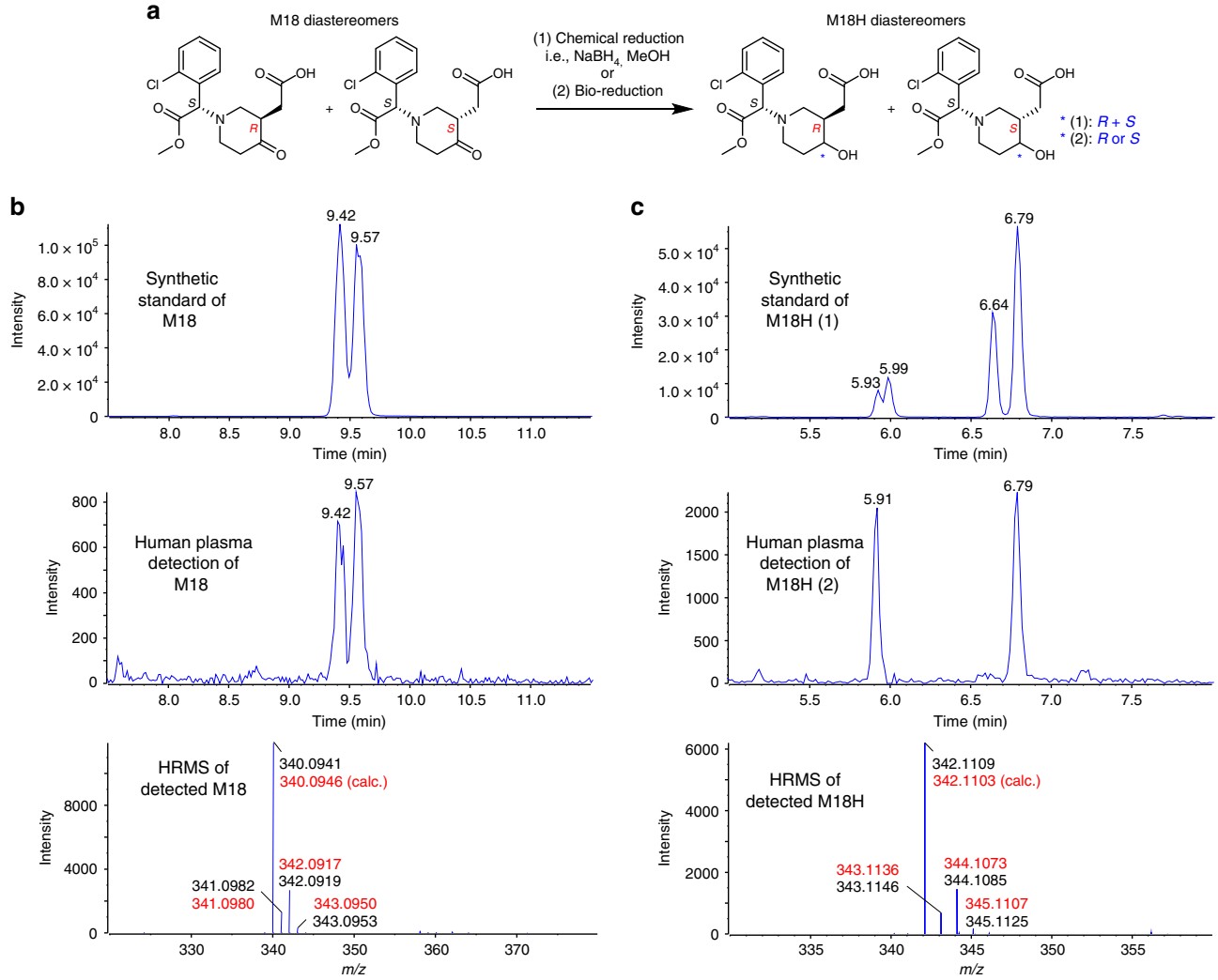

**Fig. 4** Detection of $H_2S$-released metabolites of CPG in healthy volunteers after an oral dose. **a** Structures of M18 and M18H diastereomers. Detection of **b** M18 and **c** M18H in pooled human plasma samples ($t = 1$ h, $n = 6$) by UPLC-MS/MS

monitoring together with the lack of synthetic standard have covered its $H_2S$ release pathway and therapeutic potentials. In clinical monitoring, the stabilized M15 has been found to be at similar level to the derivatized M13-H3 or M13-H4 despite of spontaneous degradation and lower detection response under target ion scan of $m/z$ 504 to $m/z$ 155 (the MS/MS of M15 is dominated by $m/z$ 212, which yields from a retro-Diels-Alder fragmentation associated with the endo structure)[36–40], the pharmacokinetics of $H_2S$ released from M15 is expected to be close to that of M13-H3 or M13-H4, which shows a $C_{max}$ of 20–40 nM with a $T_{max}$ of 1–2 h followed by an oral dose[40–43]. Given the high potency of $H_2S$ and its low endogenous concentration in human body (mostly in the nanomolar range)[4,12,57–59], the exogenous gasotransmitter released from CPG might contribute to therapeutic effects complimentary to the antiplatelet pathway of active metabolite M13-H4. Although the clinical relevance of the M15–$H_2S$ pathway to CPG therapy remains to be fully established and is beyond the focus of this manuscript, the observation that PON-1, the enzyme that catalyzes M15 formation (it might not be the only enzyme), is a genetic determinant of CPG responsiveness in certain patient populations[44–46], suggests that this pathway might be clinically important. The chemical standards of M15-DS, M15-MP, M18, and M18H yielded from our organic synthesis research can aid

future clinical investigation on this topic. Identification of $H_2S$ as the degradation product of M15 establishes CPG to be a serendipitous $H_2S$ donor that has already been widely used in clinic for over 20 years. In contrast, all rationally designed $H_2S$ donors are under either preclinical or early clinical studies seeking human use approval. Although it has been reported that some sulfhydryl-containing substances, including clinical agent zofenoprilat[60], might serve as substrates alternative to L-cysteine or L-homocysteine for the enzymatic production of $H_2S$, to our knowledge, CPG is the only one among approved drugs that releases $H_2S$ in a way irrelevant to the stringent endogenous pathways. Given that CPG has already been used in large populations as an overall safe medication, it can serve as a clinical donor probe for human studies of $H_2S$ signaling and regulation. In addition, the 20-year clinical treatment of CPG has already generated a large body of human data, which can be re-visited towards this objective.

**Model studies of masked thioenols as $H_2S$ donors**. The pathway of thiophene conversion to $H_2S$ via thiolactone and thioenol intermediates in CPG metabolism represents an interesting insight in organic chemistry, and can provide inspiration to many research areas including developing alternative $H_2S$-donating agents. Proposed in Fig. 3a, the thioketone tautomer

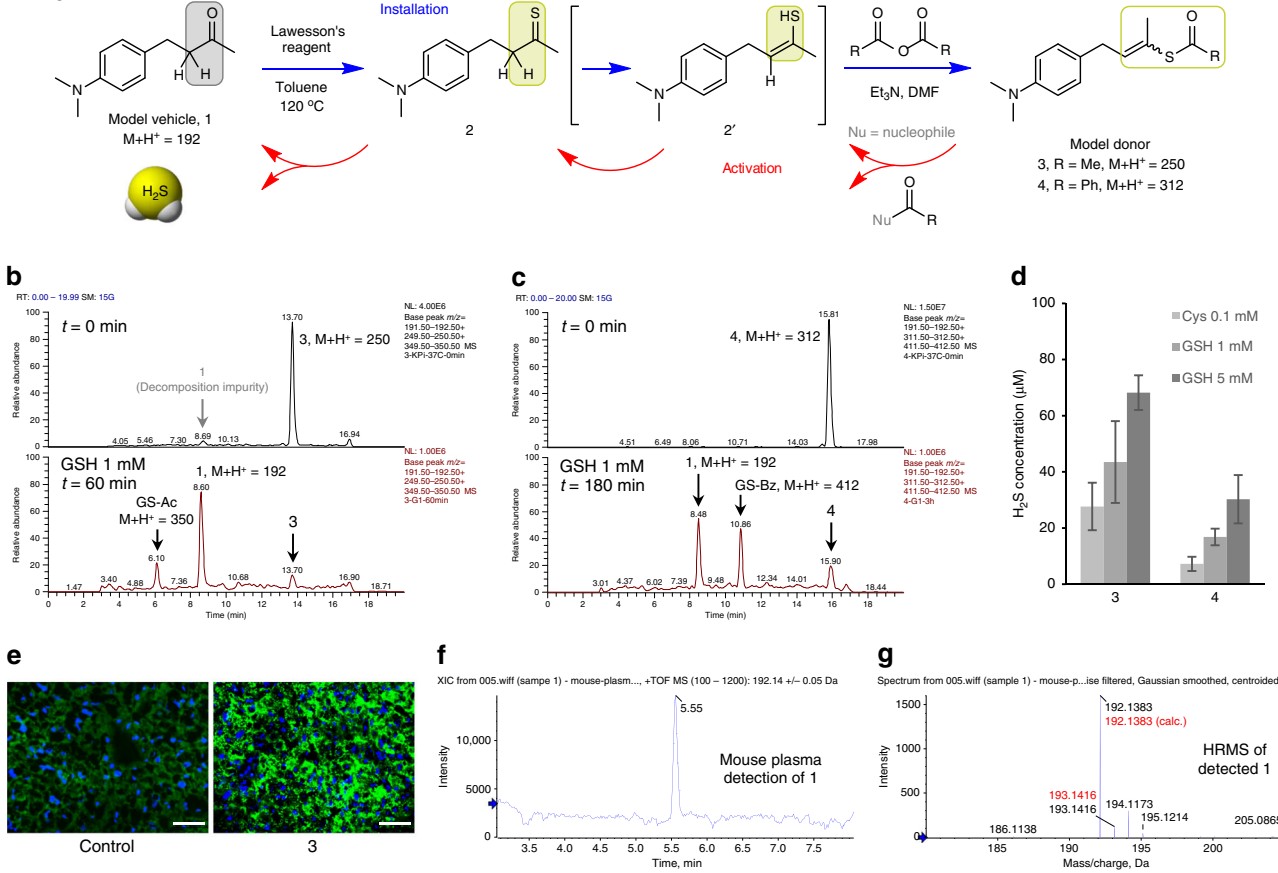

**Fig. 5** Model studies of masked thioenols as flexible $H_2S$ donors. **a** Facile preparation of $H_2S$ model donors from model vehicle **1** (blue arrows) and proposed activation through nucleophilic deacylation (red arrows). **b** In vitro activation of model donor **3** (100 μM) by GSH (1 mM, 60 min) shows recovery of model vehicle **1** and acetylated GSH (GS-Ac) in LC-MS/MS studies. **c** In vitro activation of model donor **4** (100 μM) by GSH (1 mM, 180 min) shows recovery of model vehicle **1** and benzoylated GSH (GS-Bz) in LC-MS/MS studies. **d** The $H_2S$ released from in vitro activation of model donors (100 μM) by L-cysteine (Cys, 0.1 mM) or GSH (1 or 5 mM) after 1 h was detected by methylene blue method (data is displayed as means ± S.D., $n = 3$). **e** In vivo study of $H_2S$ release from model donor **3** in mice using fluorescent imaging probe Mito-HS ($n = 4$). Scale bars are 50 μm. **f** Detection of the desulfurized metabolite (model vehicle **1**) from **3** in mouse plasma from in vivo studies ($n = 4$). **g** HRMS data of detected model vehicle **1** from mouse plasma

of thioenol metabolite M15 is a facile $H_2S$ precursor. Although its close variations such as thioamide and thioxo thioester have long been known to undergo spontaneous hydrolysis to release $H_2S$[15–17], thioketones themselves have rarely been included as $H_2S$ promoieties in recent scientific reviews and discussions. Inspired by the thioenol chemistry discovered with M15, a strategy was formulated to develop controllable $H_2S$ donors from the minimal precursor thioketo through masking its thioenol tautomer. For proof of concept, model studies were designed and conducted.

As shown in Fig. 5a, model vehicle compound (**1**) for $H_2S$ precursor assembly was designed as an aliphatic ketone fragment, i.e., 2-butanthion-4-yl, fused on a typical pharmaceutical building block, i.e., p-dimethylaminophenyl. The model ketone **1** was conveniently prepared in two steps and then sulfurized by Lawesson's reagent to yield the corresponding thioketone **2**. The thioenol tautomer of the thioketone model compound was derivatized by acetyl or benzoic anhydride to model donor **3** or **4**, respectively. In physiological buffer, both **3** and **4** have shown to be stable (Supplementary Fig. 4). Upon addition of L-glutathione (GSH), both **3** and **4** undergo deacylation to recover the thioenol structure, which tautomerizes quickly to thioketone **2** and then undergoes fast hydrolysis to recover the model vehicle compound **1** (Fig. 5b, c). In addition to GSH, model donors can also be activated by L-cysteine (Cys) and possibly other nucleophiles

under physiological conditions. The $H_2S$ released from model donor activation by thiols at physiological concentrations has been detected in vitro by the methylene blue method (Fig. 5d) and in mice by the fluorescent image probe Mito-HS (Fig. 5e). The $H_2S$-released metabolites have also been detected from the corresponding mouse plasma samples (Fig. 5f, g). It is noteworthy that model donor **3** and **4** have demonstrated different rates of thiol activation, i.e., **3** > **4** (Fig. 5d), which supports that the acyl moieties of the masked thioenol derivatives might provide tunability to $H_2S$ release rate and thiol activation specificity.

The model studies demonstrate that the $H_2S$ promoieties of masked thioenols can be conveniently installed on a minimal vehicle structure of ketone with an α-hydrogen (enolizable ketone), and the preparation sequence can be fully reversed upon bioactivation, leading to triggered $H_2S$ release and vehicle recovery, without generating much side products or additional functionalities. Since this minimal structure of enolizable ketone is widely present in clinical drugs or their metabolites (e.g. donepezil), benign substances (e.g. curcumin) and intrinsic biomolecules (e.g. testosterone), a large pool of vehicle structures are available for developing diversified $H_2S$ donors. It is interesting to note that in the CPG metabolism studies of human subjects, upon $H_2S$ release from the thioenol metabolite M15, the desulfurized ketone metabolite M18 is quickly reduced to a secondary alcohol metabolite, M18H (Fig. 4a, c). This supports

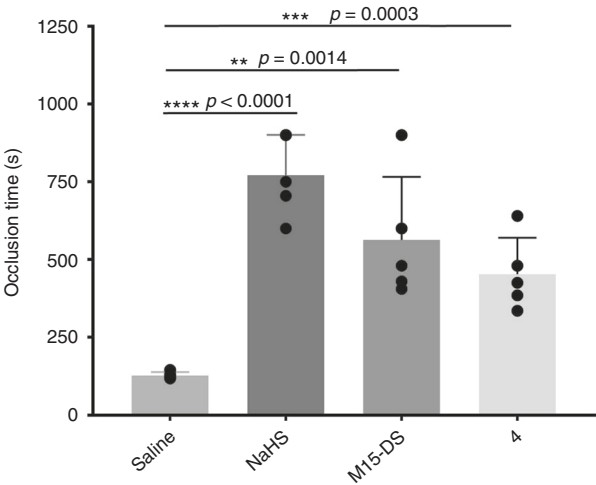

**Fig. 6** In vivo antithrombosis studies. Both the clinical H$_2$S donor and the model H$_2$S donor demonstrated inhibition to injury-induced thrombosis occlusion in mice (data is displayed as mean ± S.E.M., $n = 5$)

that in addition to the structure of ketone with an α-hydrogen, its reductive form, secondary alcohol with a β-hydrogen, can also serve as a minimal vehicle structure for conveniently installing these H$_2$S promoieties. Given that the structure of secondary alcohol with a β-hydrogen is even more widely present in clinical drugs or metabolites, benign substances and intrinsic biomolecules, the generality of this methodology can be significantly expanded. Since the vehicle molecules themselves might have well-established clinical therapeutic effects and can be fully recovered concomitant to triggered H$_2$S release, this chemical strategy can be applied to develop co-drugs of synergistic pharmacological pathways.

**In vivo antithrombosis studies of clinical donor and model donor.** The therapeutic potentials of this class of masked thioenol donors were confirmed by in vivo studies in a FeCl$_3$ carotid artery injury-induced thrombosis mouse model[61,62]. As shown in Fig. 6, upon tail vein administration of 1 mg/kg, both clinical donor M15 (in its disulfide form of M15-DS) and model donor **4** were found to significantly prolong the occlusion time of thrombosis, relative to the negative control of saline, and did so in a manner comparable to the positive control of NaHS. These results support that despite of different thioenol vehicle scaffolds in MD15-DS and **4**, H$_2$S has been released from the exogenous donors in mice, and the potent effects of H$_2$S can effectively diminish thrombosis formation and occlusion. The in vivo study result of M15-DS suggests that the circulating CPG metabolite M15 in patients might also be pharmacologically active through releasing the antithrombotic gasotransmitter H$_2$S. Uncovering this pharmacological pathway warrants future studies on calibrating the poor clinical dose–response relationship of the CPG therapy and designing personalized treatment. The observed in vivo efficacy of the model donor will stimulate research in taking the flexible promoiety of masked thioenols to a wide range of vehicle scaffolds for developing H$_2$S-donating therapeutics.

## Discussion

Thioenol represents an understudied organic structure of great biomedical potential. In this study, two basic reactivities of thioenols were demonstrated: reductivity (i.e., oxidation to disulfide) and nucleophilicity (i.e., acyl and phenacyl protection), both of which stem from the sulfhydryl group. On the other hand, the double-bond moiety endows a unique property of thioenol:

tautomerization to the H$_2$S-releasing thioketone form. This tautomerization appears to be fast with the model compound structure but much slower with the CPG metabolite: upon release in physiological buffer, thioenol form of the model compounds (**2'**, Fig. 5a) was not detected by LC-MS/MS while M15 has shown appreciable half-life. This might be due to intramolecular stabilizing effects on the thioenol structure (e.g., hydrogen bond formation with the carboxylic acid) in M15. The potential for stabilizing thioenol substructure and slowing down its tautomerization to thioketone could provide another desired tunability for enhancing the circulating half-life of the thioenol for slow and sustainable H$_2$S release.

In addition to addressing the two major challenges in H$_2$S biomedicine, the studies presented here can be extended to other directions such as: (1) re-examining other S-containing clinical agents for their potential H$_2$S-releasing metabolic pathways, (2) repurposing CPG or M15 derivatives as H$_2$S-donating agents to treat other diseases, and (3) developing the precursors of the double-bond moiety of thioenol to incorporate additional bioactivation specificity for targeted H$_2$S delivery. Although full discussion on these topics are beyond the scope of this manuscript, future explorations of these directions can be envisioned to further advance the biomedical research of H$_2$S.

In conclusion, the discovery of the H$_2$S-release pathway in CPG metabolism not only sheds light on its clinical pharmacology but also establishes it as a valuable clinical donor probe for studying H$_2$S biochemistry in human bodies. Inspired by this pathway, model studies were conducted and demonstrated that thioenol derivatives are a class of H$_2$S promoieties that can be conveniently installed on flexible vehicle molecules containing a minimal structure of ketone with an α-hydrogen (enolizable ketone) or its reductive counterpart, i.e., secondary alcohol with a β-hydrogen. The controllable and tunable bioactivation of H$_2$S release and neat recovery of the vehicle molecule can pave the avenue for developing versatile H$_2$S-donating drugs to exploit the therapeutic advantages of this important gasotransmitter. Supported by the human metabolism studies and the in vivo antithrombosis studies in mice, the results reported here can address the probe and donor challenges in H$_2$S biomedicine, and can channel a substantial advancement of the research in this field.

## Methods

**Chemical synthesis.** All commonly used chemicals were purchased from Sigma Aldrich (Milwaukee, WI) or Fisher Scientific (Pittsburgh, PA) and used without further purification. THF, acetonitrile, dichloromethane, and toluene were purified on a PPT Glass Contour 800 L Solvent System (Nashua, NH). Methyl 2-chloro-D-mandelate was purchased from TCI America (Portland, OR). Clopidogrel endo metabolite MP derivative (M15-MP) reference was purchased from Toronto Research Chemicals (Toronto, Ontario, Canada). Compound purification was achieved by flash column chromatography (SiliaFlash P60 Silica 40–63 μm 60 Å, Silicycle, Quebec City, Candada) or preparative thin-layer chromatography (Yinlong HSGF254, Yantai, China) using ethyl acetate and hexanes, or on a semi-preparative liquid chromatography (LC). The LC purification was performed on a Perkin Elmer Series 200 system (Waltham, MA). Samples were injected through a Rheodyne 7725i manual injector (Oak Harbor, WA) equipped with a 2 mL sample loop. Chromatographic separation was achieved on a Phenomenex Gemini column (100×21.20, 5.0 μm, Torrance, CA) at 25 °C using mobile phase of H$_2$O (solvent A, containing 0.1% formic acid) and MeOH (solvent B, containing 0.1% formic acid) at a flow rate of 8.0 mL/min. The UV detector was set at 254 nm, and the LC eluate was collected by a Gilson FC203B Fraction Collector (Middleton, WI). The combined factions were concentrated on rotary evaporator before lyophilized on a Labconco FreeZone 1 Liter Benchtop Freeze Dry System (Kansas City, MO).

$^1$H NMR and $^{13}$C NMR spectra were obtained on an Avance III 400 NMR (Bruker Daltonics, Billerica, MA) at 400 and 100 MHz, respectively, at ambient temperature. Chemical shifts were reported in parts per million (ppm) as referenced to residual solvent. NMR spectra were processed using MestReNova (V5.3.1, Escondido, CA). All observed protons are reported as parts per million (ppm) downfield from tetramethylsilane (TMS) or other internal reference in the appropriate solvent indicated. Data are reported as follows: chemical shift, multiplicity (s = singlet, d = doublet, t = triplet, q = quartet, br = broad,

m = multiplet), number of protons, and coupling constants. High-resolution mass spectral (HRMS) measurements were obtained on an Apex IV Fourier Transform Ion Cyclotron Resonance mass spectrometer (FT-ICR-MS, Bruker Daltonics, Billerica, MA) using a standard ESI source. The complete experimental details and compound characterization data can be found in Supplementary Information (Supplementary Fig. 1-14 and Supplementary Methods).

**LC-MS studies**. LC-MS/MS (low resolution) studies were conducted on a Thermo Surveyor HPLC system tandem a Thermo LCQ ion trap mass spectrometer (Fisher Scientific, Waltham, MA). Chromatographic separation was achieved on a Shimadzu TestKit column (50×4.6 mm, 5.0 μm, Columbia, MD) or an Agilent Zorbax C18 column (150×4.6 mm, 5.0 μm, Santa Clara, CA) at 25 °C using mobile phase of $H_2O$ (solvent A, containing 0.1% formic acid) and MeOH (solvent B, containing 0.1% formic acid) at a flow rate of 1.0 mL/min. The LC eluate was split, and 10% eluate was injected into the mass spectrometer. MS analysis was conducted using a standard electrospray ionization (ESI) source operating in positive ionization mode. Source conditions were 4.5 kV spray voltage, 225 °C heated capillary temperature, 20 V capillary voltage and sheath gas flow at 40 (arbitrary unit). The MS full scans were monitored over a mass range of $m/z$ 200 to 900. Product ion (MS/MS) scans were generated via collision-induced dissociation (CID) with helium using normalized collision energy of 60% and a precursor ion isolation width of $m/z$ 2.0. Data was centroid and processed in Qual Browser (Thermo Fisher Scientific).

UPLC-Q/TOF MS (high resolution) studies were conducted on an Acquity UPLC system (Waters, Milford, MA) tandem a Triple TOF 5600+ MS system (AB Sciex, Concord, Ontario, Canada). Chromatographic separation was achieved on an Acquity UPLC HSS T3 column (100×2.1 mm, 1.8 μm; Waters). The mobile phase consisting of $H_2O$ (solvent A, containing 5 mM ammonium acetate and 0.05% formic acid) and acetonitrile (solvent B) was delivered at a flow rate of 0.45 mL/min according to the following gradient program: 5% B for 1 min, 5% to 65% B over 14 min, 65% to 99% B over 1 min, 99% B for 1 min, 99% to 5% B over 1 min. The temperatures of the column oven and the autosampler temperatures were set at 45 °C and 4 °C, respectively. MS detection was conducted under positive ESI mode at a range of $m/z$ 100–1000. The key parameter settings include an ion spray voltage of 5500 V, a declustering potential of 60 V, an ion source heater temperature of 550 °C, a curtain gas pressure of 40 psi and ion source gas pressure of 60 psi. The collision energy for the TOF MS scans was 10 eV. For product ion scans, the collision energy was 35 eV with a spread of 10 eV. The acquisition of the MS/MS spectra was facilitated by information-dependent acquisition (IDA) including a real-time multiple mass defect filter.

**In vitro assay of $H_2S$ release**. Donor compounds (M15-DS, **3** and **4**) at different concentrations were incubated in capped flasks in the presence of corresponding activation agents (TCEP 1 mM for M15-DS; GSH 1 or 5 mM or L-cysteine 100 μM for **3** and **4**) at 37 °C. The total volume of each incubation is 1 mL. Inside the capped flasks were placed trapping wells containing 0.5 mL of 1% zinc acetate as a trapping solution and filter papers of 2.0×2.5 cm to increase the air/liquid contact surface. After 1 h, the trapping solution including the filter paper was taken out and treated by standard procedures to convert the trapped $H_2S$ to methylene blue. The absorbance of the resulting assay solution was measured at 670 nm on a Model 680 plate reader (Bio-Rad, Hercules, CA) or a PowerWave HT plate reader (BioTek, Winooski, VT). Standard curve of $H_2S$ detection was obtained with NaHS or $Na_2S$[49].

**In vivo detection of $H_2S$ release and desulfurized metabolites in mice**. Male C57BL/6 mice of 8 weeks old were used in the present study. The care and use of all animals used to generate data for this protocol was reviewed and approved by the Peking University Institutional Animal Care and Use Committee. Animals were housed in a temperature-controlled animal facility with a 12-h light/dark cycle, with water and rodent chow provided ad libitum. On the day of experiment, the animals were administered, utilizing intraperitoneal (IP) route, with saline (normal control), 26.36 mg/kg aspirin (negative reference), 5.6 mg/kg NaHS (positive reference), 13.18 mg/kg CPG (hydrogen sulfate salt, clinical donor), 13.18 mg/kg M15-DS (clinical donor), and 13.18 mg/kg **3** (model donor) by lavage. After 30 min, the animals were anesthetized using isoflurane, and blood samples were collected from angular artery. The animals were then sacrificed by cervical dislocation, and frozen liver slice samples were prepared for imaging and detection of $H_2S$. Liver slices (8 μm) were incubated with $H_2S$ imaging probe Mito-HS at a concentration of 10 μM for 1 h in dark before imaged on an A1R confocal laser-scanning microscope (Nikon Instruments, Melville, NY) with an objective lens (×60). LysoTracker Red and MitoTracker Red were used for staining lysosome and mitochondria, respectively. Emission was collected at 500–550 nm (excited at 488 nm) for green channel. LysoTracker Red and MitoTracker Red were collected at 570–620 nm (excited at 561 nm)[50,51]. For detecting the desulfurized metabolites, the drawn blood from each group of dour animals was pooled and centrifuged at 2000 × g for 5 min at 4 °C to prepare the plasma samples. For 320 μL of the pooled plasma sample, 640 μL of acetonitrile was added. The mixture was vortexed for 1 min and centrifuged at 11,000 × g for 5 min. The supernatant was evaporated to dryness and then reconstituted by 100 μL of acetonitrile and water (20:80, v/v). An aliquot of 7 μL of the resulting solution was injected to UPLC-Q/TOF MS for metabolite profiling and analysis.

**In vivo antithrombosis studies in mice**. Male C57BL/6 mice of 16 weeks old were used in the present study. The care and use of all animals used to generate data for this protocol was reviewed and approved by the Institutional Animal Care and Use Committee at the University of Texas at El Paso. All animals were housed in a temperature-controlled animal facility with a 12-h light/dark cycle, with water and rodent chow provided ad libitum. On the day of experiment, the animals were administered, utilizing intravenous (IV) tail vein route, with vehicle (saline), 1 mg/kg M15-DS (clinical donor), 1 mg/kg **4** (model donor), or 1 mg/kg NaHS (positive reference), 1 h before the occlusion time was measured. The baseline carotid artery blood flow of each animal was measured with Transonic micro-flow probe (0.5 mm, Transonic Systems Inc., Ithaca, NY) after the left carotid arteries were exposed and cleaned. Upon stabilization of blood flow, ferric chloride ($FeCl_3$, 7.5%) was applied to a filter paper disc of 1-mm diameter, which was immediately placed on top of the artery. After 3 min, the filter paper was removed, saline solution was placed in the wound, and the flow of carotid artery blood was monitored for 45 min or until it holds at zero for 2 min (stable occlusion). For the purpose of statistical analysis, 15 min was considered as the occlusion cut-off time[61,62].

**Human plasma sample preparation**. The human samples are obtained from a phase I clinical study (CTR20150346, Centre for Drug Evaluation, China Food and Drug Administration), which was approved by the Ethics Committee of Zhongshan Hospital affiliated to Fudan University (Shanghai, China). All volunteers have provided written informed consent, and the study was performed according to the principles of the Declaration of Helsinki and Good Clinical Practice. The human plasma samples were collected from six healthy Chinese subjects at 1 h after a single oral administration of 300 mg CPG. In this study, a 1.5 mL aliquot of each blood sample was drawn into an EDTA tube pretreated with a 15 μL of 3'-methoxyphenacyl bromide (MP-Br) solution (500 mM in acetonitrile) to immediately derivatize the thiol metabolites of CPG. After standing at room temperature for 10 min, the plasma samples were separated by centrifugation for 5 min at 2000 × g and stored at −70 °C. Equal volumes (50 μL) from the human plasma samples collected from the six subjects were pooled. For 50 μL of the pooled plasma sample, 150 μL of acetonitrile was added. The mixture was vortexed for 1 min and centrifuged at 11,000 × g for 5 min. The supernatant was evaporated to dryness and then reconstituted by 100 μL of acetonitrile and water (20:80, v/v). An aliquot of 7 μL of the resulting solution was injected to UPLC-Q/TOF MS for metabolite profiling and analysis.

## Data availability
The authors declare that the data supporting the findings of this study are available within the article and its Supplementary Information files, and all data are available from the authors on reasonable request.

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

## Acknowledgements

We thank Dr. A. Rettie (University of Washington), Dr. M. Kenney and Dr. C. Y. Kim (The University of Texas at El Paso) for critical reading of the manuscript, Dr. J. Tang (Peking University) for providing $H_2S$ probe Mito-HS and assisting with the fluorescent imaging study, and Prof. B. Jiao (Shandong University) and Dr. Z. Li (University of Kentucky) for attempting in vitro antiplatelet studies. Financial support was provided by the Border Biomedical Research Center (National Institutes of Health, 5G12MD007592) and the University of Texas at El Paso.

## Author contributions

Y.Z. made the initial discovery, conceived, designed, and directed the studies. Y.Z. synthesized all the compounds except M18 and performed in vitro studies of M15-DS. E.L.R. synthesized M18, purified the model donor compounds, scaled up all the synthesis, and performed in vitro studies of the model donor compounds. X.R. performed NMR analysis on M15-OMe-DS. A.J.S performed in vitro studies of $H_2S$ detection from the model donor compounds. C.D. and B.G. performed $H_2S$ detection studies of M15-DS in vitro and the donors in mice. C.L. and D.Z. performed all the in vivo metabolite identification experiments. Z.A.K., F.Z.A., and F.T.K. performed the antithrombotic experiment in mice. J.Z. performed all the high-resolution mass spectrometry analysis.

All the authors have participated in data analysis, result discussion and writing of the technical procedures. Y.Z. wrote the manuscript.

## Additional information

**Competing interests:** The authors declare no competing interests.

