## [Peer Review File · Nature Communications]

Reviewers' comments:

Reviewer #1 (Remarks to the Author):

The paper by Yaoqiu Zhu and colleagues entitled, "Enabling H₂S Biomedicine: Clopidogrel as a Clinical Donor Probe and Thioenol Derivatives as Flexible Promoieties," assesses a relatively uninvestigated metabolite of clopidogrel which contains a thioenol moiety as a donor of hydrogen sulfide in humans. In addition, the authors demonstrate the utility of novel thioenol functional groups which can be fitted to new or existing compounds to develop better H₂S probes and therapeutic agents. Their claims are generally well supported by data from in vitro, animal and human studies. The manuscript is well organized and clearly written. There are, however, some elements that require further attention before this article is accepted for publication.

Major Comments:

1. The authors highlight the numerous pathophysiological processes which are augmented by H₂S, however there is no clear explanation (or summary of current theories) of the biochemical reactions that convey its protective properties. Presumably, one of the main chemical effects would be modulation of free radicals, but this is not clear in the introductory statements. Furthermore, promising results from preclinical investigation of vitamins C and E, carotenoids, flavonoids and other antioxidants have thus far not been translated to clinical treatments. Why do the authors think H₂S supplementation via donor compounds, which presumably confers protection through similar mechanisms, would be more successful?

2. The novel thioenol conjugates outlined in the manuscript appear to not meet the authors' own criteria for successful supplementation of H₂S: "an exogenous donor needs to be bioactivated in a selective manner in response to certain stimuli" (page 2, paragraph 1). Upon exposure to glutathione the model donors undergo nonenzymatic deacylation to recover the thioenol structure which is then hydrolyzed releasing H₂S. It is unclear how this differs meaningfully from molecules which are spontaneously hydrolyzed to form H₂S such as NaHS. Please clarify in the text how these agents represent significant improvement over existing strategies.

3. Have the authors investigated whether the H₂S donating effect of clopidogrel is distinct among similar molecules? Within the same class of thienopyridine P₂Y₁₂ antagonists, prasugrel contains many similar functional groups including a thiophene moiety. Is it likely that this molecule may also participate in similar chemical processes? Prasugrel is subject to a somewhat simplified metabolic fate and might offer improved efficiency in creating the thioenol motif and generating H₂S.

4. The enzyme responsible for clopidogrel bioactivation (CYP2C19) is highly polymorphic with at least 25 variants described. Individuals of East Asian descent demonstrate the highest prevalence of loss of function genetic polymorphisms (CYP2C19*2 and *3) with more than a third of patients exhibiting markedly reduced activation of the prodrug (Scott et al., 2013; Zhu et al., 2016 and others). Were any of the individuals included in the clinical study examined for CYP2C19 status and if any were identified as "poor metabolizers" how would you explain the results in the context of your overall hypothesis? An individual's inability to activate the prodrug to the M13 metabolite would seemingly shunt the compound through alternative metabolism pathways leading to an increase in the production of the H₂S donor, M15. Yet evidence suggests that these patients are at much greater risk of thrombotic events which would indicate the protection afforded by H₂S is insufficient.

5. Some of the data presented in the manuscript (Figure 3D, specifically) do not meet the journal's guidelines for research ethics and reproducibility. Data should be presented in a way that illustrates distribution such as a box-and-whisker plot or bar graph with overlaying dot-plot of individual values. Please update to conform to established guidelines.

Minor Comments:

Spelling and grammar corrections (there may be others):

1. First page, first paragraph, third line: "much efforts" should be replaced with "great effort".
2. Second page, third paragraph, fifth line: "oxidation" should be replaced with "oxidations".
3. Second page, fourth paragraph, fourth line: "metabolite" should be replaced with "metabolites".
4. Second page, fourth paragraph, sixteenth line: "does not only cancel" should be replaced with "not only cancels".
5. Third page, first paragraph, first line: "ester" should be replaced with "esterification".
6. Fifth page, third paragraph, second line: "potentials" should be replaced with "potential".
7. Fifth page, third paragraph, sixth line: "part" should be replaced with "group".
8. Fifth page, sixth paragraph, second line: "does not only shed lights on" should be replaced with "not only sheds light on".

Reviewer #2 (Remarks to the Author):

The manuscript describes a unique approach to delivering hydrogen sulfide by using a known, FDA-approved drug. Specifically, Clopidogrel is known to be converted to a metabolic intermediate named M15, which can release H₂S and thus can be a sulfide donor. Then two other analogs (protected thioenols) were prepared and studied as sulfide donors.

There are certainly many innovative aspects and the authors should be applauded for taking on some interesting directions of finding potentially practical ways to deliver hydrogen sulfide for therapeutic applications. Given the pharmacological indications of clopidogrel and the large doses needed for efficacy as well as the known effect of hydrogen sulfide on platelet aggregation, one wonders whether clopidogrel's platelet inhibition effect is partially from hydrogen sulfide. That is to say that clopidogrel's effect on platelets may only be partially through the P2Y₁₂ ADP receptor. The significance might be more with the pharmacological mechanism than as a new hydrogen sulfide donor. The paper could be acceptable for publication if the following issues can be adequately addressed.

1. The authors state that "The maximum plasma concentration (C_{max}) of the H₂S released from bioactivated M15 after an oral dose of CPG is estimated to be in a range of 10-100 nM." Such a statement is based on indirect evidence at best, and neglects issues related to the volatility and rapid metabolism of hydrogen sulfide. Experimental evidence is needed to substantiate this point.
2. The authors state that "Given the high potency of H₂S and its low endogenous concentration in human body (mostly in the low nanomolar range)....."

Actually, the concentration of H₂S in the human body is still a controversial subject and many studies showed it to be in the high nanomolar range to low micromolar range. The reference to "low nM" is selective.

3. In Figure 2D, the X-axis label (2X) M15-DS should be clarified.
4. In Figure 2D, the methylene blue method was used to test the formation of H₂S. First, the methylene blue method is not very sensitive and won't be very accurately determine concentrations in the nanomolar range. The author is encouraged to use more accurate methods to measure H₂S production.
5. In comparing the 40-min results with the 10-h results in Figure 3C, the Model vehicle peak (The compound after H₂S release) increased a significantly, and yet the donor 4 peak barely changed. Such results are not internally consistent.
6. To support the release of H₂S from donors 3 and 4, only circumstantial evidence was provided. This is insufficient. H₂S formation needs to be directly detected and measured.
7. H₂S release from 3 and 4 in the presence of physiological concentrations of thiols (hydrogen sulfide, cysteine, and GSH) should be measured.
8. From the propose H₂S release mechanism from donors 3 and 4, thiol activation is needed (GSH). This is not new at all. It should be noted that there are several H₂S donors can release H₂S after free thiol activation; and there are many hydrolysis based H₂S donors have similar

structure. The authors need to discuss the advantage and disadvantages of their donors in the context of other donors.

9. The introduction part is very length. It is almost 2 pages.

10. For the FeCl₃ carotid artery injury-induced thrombosis mouse model test, more discussions and clarifications are needed. In addition, some kind statistical analysis is needed as well.

11. In page 4, there is statement "In physiological buffer, both 3 and 4 have shown to be stable". However, there is no stability test either in the manuscript or in supporting information document.

12. In Figure 3B, why there seems two peaks of compound 4 in the HPLC?

13. High quality of NMR spectra should be provided in supporting information. Also, among them, the NMR spectrum of compound seems to indicate impurities. Is this because of the the presence of isomers?

14. It is not clear as to what advantage there is in using clopidogrel for human indications if the goal is to treat disease because this drug already has been tested extensively. If it is to be used as a hydrogen sulfide donor, how would one de-convolute the results, give clopidogrel is a pharmacological active molecule?

Reviewer #3 (Remarks to the Author):

The authors discover and characterize an interesting new mechanism of the prodrug clopidogrel (CPG) that involves hydrogen sulfide (H₂S) release from a previously identified thioenol-containing metabolite (M15). This alternative CPG prodrug mechanism has not been appreciated before and may explain some of the unpredictable clinical outcomes associated with this drug. Insights from this novel mechanism have also been applied to the development of other H₂S donor molecules. These findings impact CPG pharmacology research, introduce new H₂S donor tool compounds, and suggest a strategy for the development of other tool compounds that may contribute to H₂S biomedicine.

The work original and should be significant to researchers in H₂S biomedicine in a number of ways. There are some issues that would need to be addressed prior to publication.

Major issues:

1) While the CPG metabolites that form after H₂S release (M18, M18H) are observed in vivo and support the proposed mechanism, the authors do not provide direct evidence for H₂S formation from either CPG in humans or M15-DS in mice. Given the complexity of the known chemistry that affects CPG in vivo, it is important to assess H₂S formation rather than the CPG metabolite byproducts, which could possibly form via different pathways. The authors should perform experiments to directly confirm that these compounds produce H₂S in vivo consistent with their in vitro studies.

2) The bar plots with numerical data (Figures 1D and 3D) are lacking information about the number of replicates performed and do not define how the data is presented (i.e. mean or median, s.d. or s.e.m., number of mice used). This information is not apparent in the text or supporting information and should be included.

3) The use of the terms "ketone- α -hydrogen" and "2°-alcohol- β -hydrogen" are confusing. Instead of "ketone- α -hydrogen", a term consistent with chemical literature is enolizable ketone, which implies that the ketone has at least one α -hydrogen substituent. Similarly, "2°-alcohol- β -hydrogen" should be changed. Perhaps including the structural representations of these groups would allow them to be clearly referenced in the text.

Minor concerns:

1) The data in figure 1D may benefit from being depicted as a scatter plot. The difference in axes scales and the y-axis break of the bar plot make the stoichiometric relationship between M15-DS and in vitro H₂S production less obvious. The x-axis label, which seems to be a reference to the 2 equivalents of M15 per molecule of M15-DS, should also be clarified in the plot and in the figure

caption.

2) In Figure 2A, the authors should specify that M18H can be generated by either chemical reduction or bio-reduction. The way the reaction is currently drawn implies that these processes are sequentially performed to prepare M18H.

Reviewers' comments:

Reviewer #1 (Remarks to the Author):

The paper by Yaoqiu Zhu and colleagues entitled, "Enabling H₂S Biomedicine: Clopidogrel as a Clinical Donor Probe and Thioenol Derivatives as Flexible Promoieties," assesses a relatively uninvestigated metabolite of clopidogrel which contains a thioenol moiety as a donor of hydrogen sulfide in humans. In addition, the authors demonstrate the utility of novel thioenol functional groups which can be fitted to new or existing compounds to develop better H₂S probes and therapeutic agents. **Their claims are generally well supported by data from in vitro, animal and human studies. The manuscript is well organized and clearly written.** There are, however, **some elements that require further attention** before this article is accepted for publication.

Major Comments:

1. The authors highlight the numerous pathophysiological processes which are augmented by H₂S, however there is no clear explanation (or summary of current theories) of the biochemical reactions that convey its protective properties. Presumably, one of the main chemical effects would be modulation of free radicals, but this is not clear in the introductory statements.

- We fully agree with the reviewer and have added a summary of H₂S biochemical reactivity in the first paragraph right after the first sentence:

"The chemical biology of H₂S are conveyed by three categories of reactions: 1) binding to the metal centers of proteins, 2) modulation of free radicals, and 3) modification of protein cysteines to persulfides.(ref 4)"

4. Filipovic, M. R., Zivanovic, J., Alvarez, B. & Banerjee, R. Chem. Rev. Chemical biology of H₂S signaling through persulfidation. 118, 1253-1337 (2018).

Furthermore, promising results from preclinical investigation of vitamins C and E, carotenoids, flavonoids and other antioxidants have thus far not been translated to clinical treatments. Why do the authors think H₂S supplementation via donor compounds, which presumably confers protection through similar mechanisms, would be more successful?

- In addition to the antioxidation effects shared with vitamin C and E, carotenoids, etc., H₂S donors can possess extra advantages in regulating proteins through modifying the cysteine residues or binding to the metal ion centers (for many metalloproteins).

- The therapeutic benefits and development prospects of H₂S donors have stimulated a wide range of research activities in this field, which have been extensively reviewed recently; we have briefly discussed it in the second paragraph and cited the most relevant review articles.

2. The novel thioenol conjugates outlined in the manuscript appear to not meet the authors' own criteria for successful supplementation of H₂S: "an exogenous donor needs to be bioactivated in a selective manner in response to certain stimuli" (page 2, paragraph 1). Upon exposure to glutathione the model donors undergo nonenzymatic deacylation to recover the thioenol structure which is then hydrolyzed releasing H₂S. It is unclear how this differs meaningfully from molecules which are spontaneously hydrolyzed to form H₂S such as NaHS. Please clarify in the text how these agents represent significant improvement over existing strategies.

- We understand and appreciate the reviewer's concern. An "ideal donor" would be activated specifically by certain stimuli to release H₂S, and "glutathione activation" represents an improvement over "spontaneous hydrolysis" towards this goal: (1) the H₂O concentration remains constant and can hardly be altered by

biological stimuli; (2) although glutathione is widely present, its concentration has been found to vary at different sites (1-10 mM) and can change in response to biological stimuli.

- In recent years, development of prodrugs (including H₂S donors) that are activated by glutathione have gained grounds, and these prodrugs are generally considered more controllable than those undergo spontaneous hydrolysis. In our manuscript, we have discussed the activation tunability of the novel donors (i.e., activation rates can be tuned against glutathione concentrations) and also the potentials to further derivative the double bond moiety of thioenol to incorporate additional activation specificity for developing more selective donors; these future research can take the novel donors of high installation flexibility to improve towards an “ideal donor”.

- In the 2nd paragraph, we have modified “an exogenous donor needs ...” to “an ideal donor needs ...”.

3. Have the authors investigated whether the H₂S donating effect of clopidogrel is distinct among similar molecules? Within the same class of thienopyridine P2Y₁₂ antagonists, prasugrel contains many similar functional groups including a thiophene moiety. Is it likely that this molecule may also participate in similar chemical processes? Prasugrel is subject to a somewhat simplified metabolic fate and might offer improved efficiency in creating the thioenol motif and generating H₂S.

- Yes, we have also been interested in studying the similar metabolic pathways of prasugrel but think it would be beyond the focus of this manuscript. In the 2nd last paragraph, we have proposed that the studies presented in this manuscript can be extended to re-examining other clinical agents for their potential H₂S-releasing metabolic pathways.

- Prasugrel was launched in 2009; although some desulfurized metabolites have been found, the endo thioenol metabolite has not been detected/established as a circulating one like the clopidogrel thioenol metabolite M15. It has been generally accepted that the simplified metabolic fate of prasugrel (e.g., no first oxidation and no methyl ester hydrolysis) significantly potentiate the putative bioactivation pathway to form the P2Y₁₂ inhibitor, and together with the much lower dose (10 mg/day), the H₂S release from prasugrel in human might be much less appreciable than from clopidogrel in clinic.

4. The enzyme responsible for clopidogrel bioactivation (CYP2C19) is highly polymorphic with at least 25 variants described. Individuals of East Asian descent demonstrate the highest prevalence of loss of function genetic polymorphisms (CYP2C19*2 and *3) with more than a third of patients exhibiting markedly reduced activation of the prodrug (Scott et al., 2013; Zhu et al., 2016 and others). Were any of the individuals included in the clinical study examined for CYP2C19 status and if any were identified as “poor metabolizers” how would you explain the results in the context of your overall hypothesis? An individual’s inability to activate the prodrug to the M13 metabolite would seemingly shunt the compound through alternative metabolism pathways leading to an increase in the production of the H₂S donor, M15. Yet evidence suggests that these patients are at much greater risk of thrombotic events which would indicate the protection afforded by H₂S is insufficient.

- We appreciate the reviewer’s discussion on the role of CYP2C19 genetic polymorphism in clopidogrel efficacy, which is a complex clinical research topic. The response of clopidogrel has been associated with high intersubject variability. Although patients of CYP2C19 loss-of-function alleles have shown reduced plasma exposure of active metabolite M13, the mechanistic role of CYP2C19 in this reduction is still not very clear, given that CYP2C19 is only one of the major CYP enzymes contributing to the two oxidative activations. There might be other biochemical mechanisms or correlations that underline this observation. For example, one of our ongoing hypotheses is that CYP2C19*2 or *3 might be able to form covalent adducts with active metabolite M13 upon its formation (catalyzed by other CYPs), leading to a diminished exposure (CYP2B6 has been found to form two types of covalent adducts with clopidogrel active metabolite. Zhang and Hollenburg, Mol. Pharmacol. 2011, 80, 839-847).

- Would a reduction in M13 activation shunt the metabolism to potentiate M15/H₂S formation? It might well if these two bioactivation pathways (leading to the formation of therapeutic agents M13 and H₂S) represent the only two or the major metabolic fates of clopidogrel. However, it has been reported that there are many heavy “metabolic attrition” pathways that accompany the metabolic activation, and additional genetic factors are involved, e.g., CYP3A4/5 and thioesterase such as PON-1 (references include a series of results from our group: ACS Med. Chem. Lett. 2012, 3, 844-849 and 2013, 4, 349-352; Chem. Res. Toxicol. 2013, 26, 179-190). All these metabolic pathways and the involved genotypes might be connected and/or tangled, which makes it challenging to predict or rationalize a clinical outcome based on one or two genetic factors.

- We are not aware of the CYP2C19 status of the healthy volunteers included in the clinical study, which is a Phase-I investigation conducted to compare a drug candidate with clopidogrel in terms of safety/tolerance profiles so the study objectives are not focused on the antithrombosis response. It would be possible to design future clinical studies to investigate this suggested topic but CYP2C19 status cannot be the only consideration. For example, if a CYP2C19 (M13 activation) low metabolizer happens to also carry a PON-1 (M15/H₂S activation) loss-of-function allele, he/she might be more likely to show low response to clopidogrel.

- We are interested in investigating on “how much does this new M15/H₂S pathway impact the response of clopidogrel”, while it is a challenging research project and is beyond the scope of this manuscript that is focused on addressing the donor probe and flexible promoiety challenges of H₂S biomedicine. We have designed some preliminary studies that will start with reconciling the PON-1 controversy, which was vigorously debated on Nature Medicine in 2011 (volume 17, page 40-41, 110-116, 1039, 1040-1041, 1042-1044).

5. Some of the data presented in the manuscript (Figure 3D, specifically) do not meet the journal’s guidelines for research ethics and reproducibility. Data should be presented in a way that illustrates distribution such as a box-and-whisker plot or bar graph with overlaying dot-plot of individual values. Please update to conform to established guidelines.

- We have modified this figure with conforming to the journal guidelines.

Minor Comments:

Spelling and grammar corrections (there may be others):

1. First page, first paragraph, third line: “much efforts” should be replaced with “great effort”.
2. Second page, third paragraph, fifth line: “oxidation” should be replaced with “oxidations”.
3. Second page, fourth paragraph, fourth line: “metabolite” should be replaced with “metabolites”.
4. Second page, fourth paragraph, sixteenth line: “does not only cancel” should be replaced with “not only cancels”.
5. Third page, first paragraph, first line: “ester” should be replaced with “esterification”.
6. Fifth page, third paragraph, second line: “potentials” should be replaced with “potential”.
7. Fifth page, third paragraph, sixth line: “part” should be replaced with “group”.
8. Fifth page, sixth paragraph, second line: “does not only shed lights on” should be replaced with “not only sheds light on”.

- We appreciate the reviewer’s help on grammars and the writing and have made all the corrections.

- The revised and finalized manuscript was carefully read by two scientific writing experts, and a few additional grammatical features were corrected to further improve the overall English and flow of the manuscript.

Reviewer #2 (Remarks to the Author):

The manuscript describes a unique approach to delivering hydrogen sulfide by using a known, FDA-approved drug. Specifically, Clopidogrel is known to be converted to a metabolic intermediate named M15, which can release H₂S and thus can be a sulfide donor. Then two other analogs (protected thioenols) were prepared and

studied as sulfide donors.

There are certainly many innovative aspects and the authors should be applauded for taking on some interesting directions of finding potentially practical ways to deliver hydrogen sulfide for therapeutic applications. Given the pharmacological indications of clopidogrel and the large doses needed for efficacy as well as the known effect of hydrogen sulfide on platelet aggregation, one wonders whether clopidogrel's platelet inhibition effect is partially from hydrogen sulfide. That is to say that clopidogrel's effect on platelets may only be partially through the P2Y₁₂ ADP receptor. The significance might be more with the pharmacological mechanism than as a new hydrogen sulfide donor. The paper could be acceptable for publication if the following issues can be adequately addressed.

1. The authors state that "The maximum plasma concentration (C_{max}) of the H₂S released from bioactivated M15 after an oral dose of CPG is estimated to be in a range of 10-100 nM." Such a statement is based on indirect evidence at best, and neglects issues related to the volatility and rapid metabolism of hydrogen sulfide. Experimental evidence is needed to substantiate this point.

- We fully agree with the reviewer on this and have modified it to:

"In clinical monitoring, the stabilized M15 has been found to be at similar level to the derivatized M13-H3 or M13-H4 despite of spontaneous degradation and lower detection response under target ion scan of "m/z 504 to m/z 155" (the MS/MS of M15 is dominated by m/z 212, which yields from a retro-Diels-Alder fragmentation associated with the endo structure), the pharmacokinetics of H₂S released from M15 is expected to be close to that of M13-H3 or M13-H4, which shows a C_{max} of 20-40 nM with a T_{max} of 1-2 hours followed by an oral dose".

- The clinical response of clopidogrel has been associated with high intersubject variability, and the reported plasma levels of activated thiol metabolites have also shown relatively high inconsistency between different clinical studies; we speculate that the H₂S release from M15 will adopt similar variability. The estimation here is to provide some general information about the level of this exogenous source of H₂S, which is appreciable compared to the endogenous source.

- The exogenous H₂S released from M15 in plasma will blend with those from endogenous production, and the overall H₂S concentration should still be under the solubility/volatility threshold; the elevated H₂S in plasma will be subjected to the same pathways of metabolism, biological interaction and regulation. Since the endogenous H₂S production, plasma level and metabolism are stringently regulated, this addition of H₂S from exogenous source might be able to pose some intervention effects.

2. The authors state that "Given the high potency of H₂S and its low endogenous concentration in human body (mostly in the low nanomolar range)...."

Actually, the concentration of H₂S in the human body is still a controversial subject and many studies showed it to be in the high nanomolar range to low micromolar range. The reference to "low nM" is selective.

- We fully agree with the reviewer on that the free H₂S concentration in human body is still a controversial topic. Actually, "how to reliably measure the free H₂S in human body" is considered as another major challenge in H₂S biomedicine in many review/perspective articles. Recent studies have shown that reaction-based H₂S measurement does not only consume free H₂S but also stimulate protein persulfides to keep releasing H₂S, leading to a measurement of "total sulfides". As the nose of a human being can sense the volatile source of 1 micromolar H₂S in water, the nanomolar range of free H₂S concentration might be a more reliable assessment. We have added three recent comprehensive review articles (Ref 4, 12, 59) that have carefully analyzed and vigorously discussed this topic and the controversy. All these references favor/support the low nanomolar assessment.

4. Filipovic, M. R., Zivanovic, J., Alvarez, B. & Banerjee, R. Chem. Rev. Chemical Biology of H₂S Signaling through Persulfidation. 118, 1253-1337 (2018).

12. Szabo, C. & Papapetropoulos, A. International union of basic and clinical pharmacology. CII: pharmacological modulation of H₂S levels: H₂S donors and H₂S biosynthesis inhibitors. *Pharmacol. Rev.* 69, 497-564 (2017).

59. Li, L., Rose, P. & Moore, P. K. Hydrogen sulfide and cell signaling. *Annu. Rev. Pharmacol. Toxicol.* 51, 169-187 (2011).

- We have modified the annotation in the parenthesis to “(mostly in the nanomolar range)” according to reviewer’s comment.

3. In Figure 2D, the X-axis label (2X) M15-DS should be clarified.

- We have modified the X-axis label to “M15”; it is the concentration of M15 after M15-DS disulfide cleavage.

4. In Figure 2D, the methylene blue method was used to test the formation of H₂S. First, the methylene blue method is not very sensitive and won’t be very accurately determine concentrations in the nanomolar range. The author is encouraged to use more accurate methods to measure H₂S production.

- We fully agree with the reviewer on that the traditional methylene blue method for trapping and detection H₂S is not very sensitive. Since it is an in vitro assessment conducted in phosphate buffer for evaluating the spontaneous degradation fate of M15 with releasing stoichiometric amount of H₂S, this method can serve the purpose.

- We have followed the reviewer’s suggestion on using more sensitive detection method (i.e. fluorescent probe) in assessing H₂S release from clopidogrel, M15-DS and model donor in the new in vivo mice experiments (requested by another reviewer).

5. In comparing the 40-min results with the 10-h results in Figure 3C, the Model vehicle peak (The compound after H₂S release) increased a significantly, and yet the donor 4 peak barely changed. Such results are not internally consistent.

- We repeated the experiments and have added the new figures that can demonstrate more clearly the GSH facilitated model donor 3 and 4 activation (nucleophilic deacylation) leading to recovery of model vehicle 1 and formation of the acylated GSH.

6. To support the release of H₂S from donors 3 and 4, only circumstantial evidence was provided. This is insufficient. H₂S formation needs to be directly detected and measured.

- As requested by the reviewer, we have added the data of direct detection of H₂S released from the model donors: in vitro in phosphate buffer by methylene blue method and in vivo in mice by fluorescence probe method (Figure 3).

7. H₂S release from 3 and 4 in the presence of physiological concentrations of thiols (hydrogen sulfide, cysteine, and GSH) should be measured.

- As requested by the reviewer, we have tested H₂S release from the model donors in the presence of physiological concentration of cysteine (range: 30-200 μM; 100 μM was tested) and GSH (range: 1-10 mM; 1 and 5 mM were tested).

- We have added the data in Figure 3 (D); we have added in the 2nd paragraph of “Model studies of masked thioenols as H₂S donors”: In addition to GSH, model donors can also be activated by L-cysteine (Cys) and possibly other nucleophiles under physiological conditions.

- Since the physiological concentration of free hydrogen sulfide remains controversial (~ nanomolar/nM) and is much lower than those of cysteine (30-200 μ M) and GSH (1-10 mM), we speculate this potential exogenous H₂S release by physiological H₂S would not be as important as the activation by GSH or cysteine.

8. From the proposed H₂S release mechanism from donors 3 and 4, thiol activation is needed (GSH). This is not new at all. It should be noted that there are several H₂S donors that can release H₂S after free thiol activation; and there are many hydrolysis based H₂S donors that have similar structure. The authors need to discuss the advantages and disadvantages of their donors in the context of other donors.

- We fully agree with the reviewer's comment, and this is why in the 3rd paragraph (introduction of H₂S donors) of the manuscript we have discussed the many reported donors that are activated by GSH. We have also discussed the donors that undergo spontaneous hydrolysis in the same paragraph. As reviewed in recent articles (cited in this paragraph), compared to spontaneous hydrolysis, GSH attack represents a more controllable activation: (1) the H₂O concentration remains constant and can hardly be altered by biological stimuli; (2) although glutathione is widely present, its concentration has been found to vary at different sites (1-10 mM) and can change in response to biological stimuli.

- In the manuscript, we have demonstrated the high installation flexibility of the novel donors as its major advantage; we have also discussed the activation tunability and the potentials to further derivative the double bond moiety of thioenol to incorporate additional activation specificity for developing more selective and controllable donors.

9. The introduction part is very long. It is almost 2 pages.

- We have edited and deleted some sentences in the introduction part according to reviewer's comment; however another reviewer has requested to add some background information about the H₂S biochemical reactions. Since the manuscript covers a relatively wide field including clopidogrel pharmacology, H₂S clinical donor probe and model studies of new H₂S promoiety, the background and introduction part would serve as the informative foundation to connect and organize these parts to build a meaningful content for tackling the biomedical challenges.

- The introduction part contains 668 words, which meets the journal requirement (less than 1,000 words).

10. For the FeCl₃ carotid artery injury-induced thrombosis mouse model test, more discussions and clarifications are needed. In addition, some kind statistical analysis is needed as well.

- As requested by the reviewer, we have added additional discussion in the "In vivo antithrombosis studies of clinical donor and model donor" paragraph on the FeCl₃ carotid artery injury-induced thrombosis mouse model test:

"These results support that despite of different thioenol vehicle scaffolds in MD15-DS and 4, H₂S has been released from the exogenous donors in mice, and the potent effects of H₂S can effectively diminish thrombosis formation and occlusion. The in vivo study result of M15-DS suggests that the circulating CPG metabolite M15 in patients might also be pharmacologically active through releasing the antithrombotic gasotransmitter H₂S. Uncovering this new pharmacological pathway warrants future studies on calibrating the poor clinical dose-response relationship of the CPG therapy and designing personalized treatment. The observed in vivo efficacy of the model donor will stimulate research in taking the flexible promoiety of masked thioenols to a wide range of vehicle scaffolds for developing novel H₂S therapeutics."

- We have modified the result/figure with conforming to the journal guidelines on statistical analysis.

11. In page 4, there is statement "In physiological buffer, both 3 and 4 have shown to be stable". However, there is no stability test either in the manuscript or in supporting information document.

- We have added stability study results in Supplementary Information as requested by the reviewer.
- We have modified this sentence to: In physiological buffer, both 3 and 4 have shown to be stable (Supplementary Fig. 4).

12. In Figure 3B, why there seems two peaks of compound 4 in the HPLC?

- As shown in Figure 3A, the preparation of the acylated thioenols result in very close double bond Z/E isomers (not at equal ratios), which can be reflected by the LC-MS peaks on certain LC columns and/or LC programs. Since the activation will lead to tautomerization/cancellation of this double bond, we did not separate or differentiate these isomers.
- We repeated the LC-MS/MS studies with another LC column, and the very close double bond Z/E isomers of 4 show up as a symmetric peak.

13. High quality of NMR spectra should be provided in supporting information. Also, among them, the NMR spectrum of compound seems to indicate impurities. Is this because of the presence of isomers?

- We have reformatted the NMR spectra to full page sizes in Supplementary Information. As explained above, some spectra reflect the presence of isomers.

14. It is not clear as to what advantage there is in using clopidogrel for human indications if the goal is to treat disease because this drug already has been tested extensively. If it is to be used as a hydrogen sulfide donor, how would one de-convolute the results, give clopidogrel is a pharmacological active molecule?

- We appreciate the reviewer's comments and discussion. The clinical response of clopidogrel has been associated with high intersubject variability that can be only partially explained or predicted by known pathways and the involved genotypes. The new M15/H₂S pathway might provide a much needed calibration to assess the clinical outcome and design personalized treatment.
- The hydrogen sulfide released from clopidogrel can pose a series of beneficial effects: 1) vasodilation (different from active metabolite), 2) cardiovascular homeostasis (different from active metabolite), and 3) antiplatelet via P2Y₁₂ inhibition (same as active metabolite). For the same antiplatelet effect, while active metabolite directly antagonizes the platelet aggregation receptor P2Y₁₂, H₂S proceeds with a different mechanism involving the NOS-CD31 signaling pathway (Li, et al. Int. J. Clin. Exp. Med. 2016, 9, 15607-15620), which might also be characterized by cytosolic Ca²⁺ mobilization and/or cAMP-dependence (Nishikawa, et al. 2013, Biol. Pharm. Bull. 36, 1278-1282). We are interested in working with collaborators in future on these research topics suggested by the reviewer.

Reviewer #3 (Remarks to the Author):

The authors discover and characterize an interesting new mechanism of the prodrug clopidogrel (CPG) that involves hydrogen sulfide (H₂S) release from a previously identified thioenol-containing metabolite (M15). This alternative CPG prodrug mechanism has not been appreciated before and may explain some of the unpredictable clinical outcomes associated with this drug. Insights from this novel mechanism have also been applied to the development of other H₂S donor molecules. These findings impact CPG pharmacology research, introduce new H₂S donor tool compounds, and suggest a strategy for the development of other tool compounds that may contribute to H₂S biomedicine.

The work (is) original and should be significant to researchers in H₂S biomedicine in a number of ways. There are some issues that would need to be addressed prior to publication.

Major issues:

1) While the CPG metabolites that form after H₂S release (M18, M18H) are observed in vivo and support the proposed mechanism, the authors do not provide direct evidence for H₂S formation from either CPG in humans or M15-DS in mice. Given the complexity of the known chemistry that affects CPG in vivo, it is important to assess H₂S formation rather than the CPG metabolite byproducts, which could possibly form via different pathways. The authors should perform experiments to directly confirm that these compounds produce H₂S in vivo consistent with their in vitro studies.

- We appreciate the reviewer's suggestion and have conducted the requested new experiment of detecting H₂S release in vivo in mice upon intraperitoneal (IP) administration of CPG and M15-DS. NaHS was used as the positive control, and aspirin (a known antiplatelet agent devoid of H₂S release) was used as the negative control. The sensitive fluorescent probe method was adopted to analyze H₂S in mice liver slices. Evident fluorescence was observed in the CPG group and the M15-DS group, which supports that H₂S is produced from these exogenous donors. We have included the fluorescence data in Figure 1 (as 1E), and have added the study design, result/discussion and experimental methods in the manuscript.

- We have also analyzed the plasma samples from the above new in vivo studies by UPLC-MS/MS and detected the corresponding desulfurized metabolites M18 (and M18H) (Figure 1F). These results further support that M18/M18H can serve as the marker for H₂S release. We have added these study results.

2) The bar plots with numerical data (Figures 1D and 3D) are lacking information about the number of replicates performed and do not define how the data is presented (i.e. mean or median, s.d. or s.e.m., number of mice used). This information is not apparent in the text or supporting information and should be included.

- We have modified these figures with conforming to the journal guidelines on statistical analysis.

3) The use of the terms "ketone- α -hydrogen" and "2°-alcohol- β -hydrogen" are confusing. Instead of "ketone- α -hydrogen", a term consistent with chemical literature is enolizable ketone, which implies that the ketone has at least one α -hydrogen substituent. Similarly, "2°-alcohol- β -hydrogen" should be changed. Perhaps including the structural representations of these groups would allow them to be clearly referenced in the text.

- We understand the reviewer's concern and appreciate the suggestion. We have changed the term of "ketone- α -hydrogen" to "ketone with an α -hydrogen (enolizable ketone)". Similarly, we have changed the term of "2°-alcohol- β -hydrogen" to "secondary alcohol with a β -hydrogen".

- The representation of these substructures, e.g., CH-C(=O)-, CH-CH(OH)-, might not be able to convey the structural information clearly and are thus not included.

Minor concerns:

1) The data in figure 1D may benefit from being depicted as a scatter plot. The difference in axes scales and the y-axis break of the bar plot make the stoichiometric relationship between M15-DS and in vitro H₂S production less obvious. The x-axis label, which seems to be a reference to the 2 equivalents of M15 per molecule of M15-DS, should also be clarified in the plot and in the figure caption.

- We appreciate the reviewer's suggestion. Since Figure 1D can clearly reflect the cleaved M15 releases (close to) stoichiometric amount of H₂S, we decided not to reformat it.

- We have modified the X-axis label to "M15"; it is the concentration of M15 after M15-DS disulfide cleavage.

2) In Figure 2A, the authors should specify that M18H can be generated by either chemical reduction or bio-reduction. The way the reaction is currently drawn implies that these processes are sequentially performed to prepare M18H.

- We fully agree with the reviewer that this drawing is confusing and have modified it: an "or" was added between the two conditions, and only one arrow was used (under the conditions).

REVIEWERS' COMMENTS:

Reviewer #1 (Remarks to the Author):

The paper by Yaoqiu Zhu and colleagues entitled, "Enabling H₂S Biomedicine: Clopidogrel as a Clinical Donor Probe and Thioenol Derivatives as Flexible Promoieties," assesses a relatively uninvestigated metabolite of clopidogrel which contains a thioenol moiety as a donor of hydrogen sulfide in humans. In addition, the authors demonstrate the utility of novel thioenol functional groups which can be fitted to new or existing compounds to develop better H₂S probes and therapeutic agents. Thank you for responding to my comments and for your careful revision of the manuscript. The revised version satisfactorily addresses my concerns and clarifies some of the authors interpretation of the results. Based on these changes, I feel that this article is now acceptable for publication.

Reviewer #2 (Remarks to the Author):

The authors have satisfactorily addressed the concerns of this reviewer. The manuscript is now acceptable for publication.

Reviewer #3 (Remarks to the Author):

The authors have addressed the issues that I had raised in the initial review. I have no further concerns and am satisfied with the revised manuscript.

Reviewer #1 (Remarks to the Author):

The paper by Yaoqiu Zhu and colleagues entitled, "Enabling H₂S Biomedicine: Clopidogrel as a Clinical Donor Probe and Thioenol Derivatives as Flexible Promoieties," assesses a relatively uninvestigated metabolite of clopidogrel which contains a thioenol moiety as a donor of hydrogen sulfide in humans. In addition, the authors demonstrate the utility of novel thioenol functional groups which can be fitted to new or existing compounds to develop better H₂S probes and therapeutic agents. Thank you for responding to my comments and for your careful revision of the manuscript. The revised version satisfactorily addresses my concerns and clarifies some of the authors interpretation of the results. Based on these changes, I feel that this article is now acceptable for publication.

Reviewer #2 (Remarks to the Author):

The authors have satisfactorily addressed the concerns of this reviewer. The manuscript is now acceptable for publication.

Reviewer #3 (Remarks to the Author):

The authors have addressed the issues that I had raised in the initial review. I have no further concerns and am satisfied with the revised manuscript.

- We truly appreciate all the three reviewers for providing their expert comments, opinions and suggestions on helping us improve the manuscript.